# Expressing and Exploiting the Common Subgoal Structure
# of Classical Planning Domains Using Sketches

**Dominik Drexler[1], Jendrik Seipp[1], Hector Geffner[3,2,1]**

[1]Linköping University, Linköping, Sweden
[2]Universitat Pompeu Fabra, Barcelona, Spain
[3]Institució Catalana de Recerca i Estudis Avançats (ICREA), Barcelona, Spain
{dominik.drexler, jendrik.seipp}@liu.se, hector.geffner@upf.edu

## Abstract

Width-based planning methods deal with conjunctive goals by decomposing problems into subproblems of low width. Algorithms like SIW thus fail when the goal is not easily serializable in this way or when some of the subproblems have a high width. In this work, we address these limitations by using a simple but powerful language for expressing finer problem decompositions introduced recently by Bonet and Geffner, called *policy sketches*. A policy sketch $R$ over a set of Boolean and numerical features is a set of sketch rules $C \mapsto E$ that express how the values of these features are supposed to change. Like general policies, policy sketches are domain general, but unlike policies, the changes captured by sketch rules do not need to be achieved in a single step. We show that many planning domains that cannot be solved by SIW are provably solvable in low polynomial time with the $SIW_R$ algorithm, the version of SIW that employs user-provided policy sketches. Policy sketches are thus shown to be a powerful language for expressing domain-specific knowledge in a simple and compact way and a convenient alternative to languages such as HTNs or temporal logics. Furthermore, they make it easy to express general problem decompositions and prove key properties of them like their width and complexity.

## Introduction

The success of width-based methods in classical planning is the result of two main ideas: the use of conjunctive goals for decomposing a problem into subproblems, and the observation that the width of the subproblems is often bounded and small (Lipovetzky and Geffner 2012). When these assumptions do not hold, pure width-based methods struggle and need to be extended with heuristic estimators or landmark counters that yield finer problem decompositions (Lipovetzky and Geffner 2017a,b). These hybrid approaches have resulted in state-of-the-art planners but run into shortcomings of their own: unlike pure width-based search methods, they require declarative, PDDL-like action models and thus cannot plan with black box simulators (Lipovetzky, Ramirez, and Geffner 2015; Shleyfman, Tuisov, and Domshlak 2016; Geffner and Geffner 2015), and they produce decompositions that are ad-hoc and difficult to understand. Variations of these approaches, where the use of declarative action models is replaced by polynomial searches, have pushed the scope of pure-width based search methods (Francès et al.

2017), but they do not fully overcome their basic limits: *top goals that are not easily serializable or that have a high width*. These are indeed the limitations of one of the simplest width-based search methods, Serialized Iterated Width (SIW) that greedily achieves top goal first, one at a time, using IW searches (Lipovetzky and Geffner 2012).

In this work, we address the limitations of the SIW algorithm differently by using a simple but powerful language for expressing richer problem decompositions recently introduced by Bonet and Geffner (2021), called **policy sketches**. A policy sketch is a set of sketch rules over a set of Boolean and numerical features of the form $C \mapsto E$ that express how the values of the features are supposed to change. Like **general policies** (Bonet and Geffner 2018), sketches are general and not tailored to specific instances of a domain, but unlike policies, the feature changes expressed by sketch rules represent subgoals that do not need to be achieved in a single step.

We pick up here where Bonet and Geffner left off and show that many benchmark planning domains that SIW cannot solve are provably solvable in low polynomial time through the $SIW_R$ algorithm, the version of SIW that makes use of user-provided policy sketches. Policy sketches are thus shown to be a powerful **language for expressing domain-specific knowledge** in a simple and compact way and a convenient alternative to languages such as HTNs or temporal logics. Bonet and Geffner introduce the language of sketches and the theory behind them; we show their use and the properties that follow from them. As we will see, unlike HTNs and temporal logics, sketches can be used to **express and exploit the common subgoal structure of planning domains** without having to express how subgoals are to be achieved. Also, by being simple and succinct they provide a convenient target language for learning the subgoal structure of domains automatically, although this problem, related to the problem of learning general policies (Bonet, Francès, and Geffner 2019; Francès, Bonet, and Geffner 2021), is outside the scope of this paper. In this work, we use sketches to solve domains in polynomial time, which excludes intractable domains. Indeed, intractable domains do not have general policies nor sketches of bounded width and require non-polynomial searches. Sketches and general policies, however, are closely related: sketches provide the skeleton of a general policy, or a general policy with "holes"

that are filled by searches that can be shown to be polynomial (Bonet and Geffner 2021).

The paper is organized as follows. We review the notions of width, sketch width, and policy sketches following Bonet and Geffner (2021). We show then that it is possible to write compact and transparent policy sketches for many domains, establish their widths, and analyze the performance of the SIW$_R$ algorithm. We compare sketches to HTNs and temporal logics, briefly discuss the challenge of learning sketches automatically, and summarize the main contributions.

## Planning and Width

A *classical planning problem* or instance $P = (D, I)$ is assumed to be given by a first-order domain $D$ with action schemas defined over some domain predicates, and instance information $I$ describing a set of objects, and two sets of ground literals describing the initial situation $Init$ and goal description $Goal$. The initial situation is assumed to be complete such that either $L$ or its complement is in $Init$. A problem $P$ defines a state model $S(P) = (S, s_0, G, Act, A, f)$ where the states in $S$ are the truth valuations over the ground atoms represented by the set of literals that they make true, the initial state $s_0$ is $Init$, the set of goal states $G$ includes all of those that make the goal atoms in $Goal$ true, the actions $Act$ are the ground actions obtained from the action schemas and the objects, the actions $A(s)$ applicable in state $s$ are those whose preconditions are true in $s$, and the state transition function $f$ maps a state $s$ and an action $a \in A(s)$ into the successor state $s' = f(a, s)$. A *plan* $\pi$ for $P$ is a sequence of actions $a_0, \ldots, a_n$ that is executable in $s_0$ and maps the initial state $s_0$ into a goal state; i.e., $a_i \in A(s_i)$, $s_{i+1} = f(a_i, s_i)$, and $s_{n+1} \in G$. The cost of a plan is assumed to be given by its length, and a plan is optimal if there is no shorter plan. We'll be interested in solving *collections* of well-formed instances $P = (D, I)$ over fixed domains $D$ denoted as $\mathcal{Q}_D$ or simply as $\mathcal{Q}$.

The most basic width-based search method for solving a planning problem $P$ is IW(1). It performs a standard breadth-first search in the rooted directed graph associated with the state model $S(P)$ with one modification: IW(1) prunes a newly generated state if it does not make an atom $r$ true for the first time in the search. The procedure IW($k$) for $k > 1$ is like IW(1) but prunes a state if a newly generated state does not make a collection of up to $k$ atoms true for the first time. Underlying the IW algorithms is the notion of *problem width* (Lipovetzky and Geffner 2012):

**Definition 1** (Width). *Let $P$ be a classical planning problem with initial state $s_0$ and goal states $G$. The **width** $w(P)$ of $P$ is the minimum $k$ for which there exists a **sequence** $t_0, t_1, \ldots, t_m$ **of atom tuples** $t_i$, each consisting of at most $k$ atoms, such that:*

1. *$t_0$ is true in initial state $s_0$ of $P$,*
2. *any optimal plan for $t_i$ can be extended into an optimal plan for $t_{i+1}$ by adding a single action, $i = 1, \ldots, n-1$,*
3. *if $\pi$ is an optimal plan for $t_m$, then $\pi$ is an optimal plan for $P$.*

If a problem $P$ is unsolvable, $w(P)$ is set to the number of variables in $P$, and if $P$ is solvable in at most one step,

$w(P)$ is set to 0 (Bonet and Geffner 2021). Chains of tuples $\theta = (t_0, t_1, \ldots, t_m)$ that comply with conditions 1–3 are called **admissible**, and the size of $\theta$ is the size $|t_i|$ of the largest tuple in the chain. We talk about the third condition by saying that $t_m$ implies $G$ in the admissible chain $t_1, t_2, \ldots, t_m$. The width $w(P)$ is thus $k$ if $k$ is the minimum size of an admissible chain for $P$. If the width of a problem $P$ is $w(P) = k$, IW($k$) finds an optimal (shortest) plan for $P$ in time and space that are exponential in $k$ and not in the number of problem variables $N$ as breadth-first search.

The IW($k$) algorithm expands up to $N^k$ nodes, generates up to $bN^k$ nodes, and runs in time and space $O(bN^{2k-1})$ and $O(bN^k)$, respectively, where $N$ is the number of atoms and $b$ bounds the branching factor in problem $P$. IW($k$) is guaranteed to solve $P$ optimally (shortest path) if $w(P) \leq k$. If the width of $P$ is not known, the **IW** algorithm can be run instead which calls IW($k$) iteratively for $k = 0, 1, \ldots, N$ until the problem is solved, or found to be unsolvable.

While IW and IW($k$) algorithms are not practical by themselves, they are building blocks for other methods. Serialized Iterated Width or **SIW** (Lipovetzky and Geffner 2012), starts at the initial state $s = s_0$ of $P$, and then performs an IW search from $s$ to find a shortest path to state $s'$ such that $\#g(s') < \#g(s)$ where $\#g$ counts the number of top goals of $P$ that are not true in $s$. If $s'$ is not a goal state, $s$ is set to $s'$ and the loop repeats until a goal state is reached.

In practice, the IW($k$) searches in SIW are limited to $k \leq 2$ or $k \leq 3$, so that SIW solves a problem or fails in low polynomial time. SIW performs well in many benchmark domains but fails in problems where the width of some top goal is not small, or the top goals can't be serialized greedily. More recent methods address these limitations by using width-based notions (novelty measures) in complete best-first search algorithms (Lipovetzky and Geffner 2017a; Francès et al. 2017), yet they also struggle in problems where some top goals have high width. In this work, we take a different route: we keep the greedy polynomial searches underlying SIW but consider a richer class of problem decompositions expressed through sketches. The resulting planner SIW$_R$ is **not** domain-independent like SIW, but it illustrates that a bit of domain knowledge can go a long way in the effective solution of arbitrary domain instances.

## Features and Sketches

A **feature** is a function of the state over a class of problems $\mathcal{Q}$. The features considered in the language of sketches are Boolean, taking values in the Boolean domain, or numerical, taking values in the non-negative integers. For a set $\Phi$ of features and a state $s$ of some instance $P$ in $\mathcal{Q}$, $f(s)$ is the **feature valuation** determined by a state $s$. A Boolean feature valuation over $\Phi$ refers instead to the valuation of the expressions $p$ and $n = 0$ for Boolean and numerical features $p$ and $n$ in $\Phi$. If $f$ is a feature valuation, $b(f)$ will denote the Boolean feature valuation determined by $f$ where the values of numerical features are just compared with 0.

The set of features $\Phi$ **distinguish** or **separate the goals** in $\mathcal{Q}$ if there is a set $B_{\mathcal{Q}}$ of Boolean feature valuations such that $s$ is a goal state of an instance $P \in \mathcal{Q}$ iff $b(f(s)) \in B_{\mathcal{Q}}$. For example, if $\mathcal{Q}_{clear}$ is the set of all blocks world instances

with stack/unstack operators and common goal $clear(x) \land handempty$ for some block $x$, and $\Phi = \{n(x), H\}$ are the features that track the number of blocks above $x$ and whether the gripper is holding a block, then there is a single Boolean goal valuation that makes the expression $n(x) = 0$ true and $H$ false.

A **sketch rule** over features $\Phi$ has the form $C \mapsto E$ where $C$ consists of Boolean feature conditions, and $E$ consists of feature effects. A Boolean (feature) condition is of the form $p$ or $\neg p$ for a Boolean feature $p$ in $\Phi$, or $n = 0$ or $n > 0$ for a numerical feature $n$ in $\Phi$. A feature effect is an expression of the form $p$, $\neg p$, or $p?$ for a Boolean feature $p$ in $\Phi$, and $n\downarrow$, $n\uparrow$, or $n?$ for a numerical feature $n$ in $\Phi$. The syntax of sketch rules is the syntax of the policy rules used to define generalized policies (Bonet and Geffner 2018), but their semantics is different. In policy rules, the effects have to be delivered in one step by state transitions, while in sketch rules, they can be delivered by longer state sequences.

A **policy sketch** $R_\Phi$ is a collection of sketch rules over the features $\Phi$ and the sketch is well-formed if it is built from features that **distinguish the goals** in $\mathcal{Q}$, and is **terminating** (to be made precise below). A **well-formed sketch** for a class of problems $\mathcal{Q}$ defines a serialization over $\mathcal{Q}$; namely, a "preference" ordering '$\prec$' over the feature valuations that is irreflexive and transitive, and which is given by the smallest strict partial order that satisfies $f' \prec f$ if $f'$ is not a goal valuation and the pair of feature valuations $(f, f')$ **satisfies a sketch rule** $C \mapsto E$. This happens when: 1) $C$ is true in $f$, 2) the Boolean effects $p$ ($\neg p$) in $E$ are true in $f'$, 3) the numerical effects are satisfied by the pair $(f, f')$; i.e., if $n\downarrow$ in $E$ (resp. $n\uparrow$), then the value of $n$ in $f'$ is smaller than in $f$, i.e., $f'_n < f_n$ (resp. $f_n > f'_n$), and 4) Features that do not occur in $E$ have the same value in $f$ and $f'$. Effects $p?$ and $n?$ do not constraint the value of the features $p$ and $n$ in any way, and by including them in $E$, we say that they can change in any way, as opposed to features that are not mentioned in $E$ whose values in $f$ and $f'$ must be the same.

Following Bonet and Geffner, we do not use the serializations determined by sketches but their associated problem **decompositions**. The set of **subgoal states** $G_r(s)$ associated with a sketch rule $r : C \mapsto E$ in $R_\Phi$ and a state $s$ for a problem $P \in Q$, is empty if $C$ is not true in $f(s)$, and else is given by the set of states $s'$ with feature valuations $f(s')$ such that the pair $(f, f')$ for $f = f(s)$ and $f' = f(s')$ satisfies the sketch rule $r$. Intuitively, when in a state $s$, the subgoal states $s'$ in $G_r(s)$ provide a stepping stone in the search for plans connecting $s$ to the goal of $P$.

## Serialized Iterated Width with Sketches

The **SIW$_R$** algorithm is a variant of the SIW algorithm that uses a given sketch $R = R_\Phi$ for solving problems $P \in \mathcal{Q}$. SIW$_R$ starts at the state $s := s_0$, where $s_0$ is the initial state of $P$, and then performs an IW search to find a state $s'$ that is closest from $s$ such that $s'$ is a goal state of $P$ or a subgoal state in $G_r(s)$ for some sketch rule $r$ in $R$. If $s'$ is not a goal state, then $s$ is set to $s'$, $s := s'$, and the loop repeats until a goal state is reached. The features define subgoal states through the sketch rules but otherwise play no role in the IW searches.

The only difference between SIW and SIW$_R$ is that in SIW each IW search finishes when the goal counter $\#g$ is decremented, while in SIW$_R$, when a goal or subgoal state is reached. The behavior of plain SIW can be emulated in SIW$_R$ using the single sketch rule $\{\#g > 0\} \mapsto \{\#g\downarrow\}$ in $R$ when the goal counter $\#g$ is the only feature, and the rule $\{\#g > 0\} \mapsto \{\#g\downarrow, p?, n?\}$, when $p$ and $n$ are the other features. This last rule says that it is always "good" to decrease the goal counter independently of the effects on other features, or alternatively, that decreasing the goal counter is a subgoal from any state $s$ where $\#g(s)$ is positive.

The complexity of SIW$_R$ over a class of problems $\mathcal{Q}$ can be bounded in terms of the **width of the sketch** $R_\Phi$, which is given by the width of the possible subproblems that can be encountered during the execution of SIW$_R$ when solving a problem $P$ in $\mathcal{Q}$. For this, let us define the set $S_R(P)$ of reachable states in $P$ when following the sketch $R = R_\Phi$ recursively as follows: 1) the initial state $s$ of $P$ is in $S_R(P)$, 2) the (subgoal) states $s' \in G_r(s)$ that are closest to $s$ are in $S_R(P)$ if $s \in S_R(P)$ and $r \in R$. The states in $S_R(P)$ are called the $R$-reachable states in $P$. The width of the sketch $R$ is then (Bonet and Geffner 2021):

**Definition 2** (Sketch width). *The width of the sketch $R = R_\Phi$ at state $s$ of problem $P \in \mathcal{Q}$, denoted $w_R(P[s])$, is the width $k$ of the subproblem $P[s]$ that is like $P$ but with initial state $s$ and goal states that contain those of $P$ and those in $G_r(s)$ for all $r \in R$. The **width of the sketch** $R$ over $\mathcal{Q}$ is $w_R(\mathcal{Q}) = \max_{P,s} w_R(P[s])$ for $P \in \mathcal{Q}$ and $s \in S_R(P)$.*[1]

The time complexity of SIW$_R$ can then be expressed as follows, under the assumption that the features are all linear (Bonet and Geffner 2021):

**Theorem 1.** *If width $w_R(\mathcal{Q}) = k$, SIW$_{R_\Phi}$ solves any $P \in \mathcal{Q}$ in $O(bN^{|\Phi|+2k-1})$ time and $O(bN^k + N^{|\Phi|+k})$ space.*

A feature is linear if it can be computed in linear time and can take a linear number of values at most. In both cases, the linearity is in the number of atoms $N$ in the problem $P$ in $\mathcal{Q}$. If the features are not linear but polynomial in $P$, the bounds on SIW$_R$ remain polynomial as well (both $k$ and $\Phi$ are constants).

Bonet and Geffner introduce and study the language of sketches as a variation of the language of general policies and their relation to the width and serialized width of planning domains. They illustrate the use of sketches in a simple example (Delivery) but focus mainly on the theoretical aspects. Here we focus instead on their use for modeling domain-specific knowledge in the standard planning benchmarks as an alternative to languages like HTNs.

## Sketches for Classical Planning Domains

In this section, we present policy sketches for a representative set of classical planning domains from the benchmark

---

[1]This definition changes the one by Bonet and Geffner slightly by restricting the reachable states $s$ to those that are $R$-reachable; i.e., part of $S_R(P)$. This distinction is convenient when $\mathcal{Q}$ does not contain all possible "legal" instances $P$ but only those in which the initial situations complies with certain conditions (e.g., robot arm is empty). In those cases, the sketches for $\mathcal{Q}$ do not have to cover all reachable states.

set of the International Planning Competition (IPC). All of the chosen domains are solvable suboptimally in polynomial time, but plain SIW fails to solve them. There are two main reasons why SIW fails. First, if achieving a single goal atom already has a sufficiently large width. Last, greedy goal serialization generates such avoidable subproblems, including reaching unsolvable states.

We provide a handcrafted sketch for each of the domains and show that it is well-formed and has small sketch width. These sketches allow SIW$_R$ to solve all instances of the domain in low polynomial time and space by Theorem 1. Furthermore, we impose a low polynomial complexity bound on each feature, i.e., at most quadratic in the number of grounded atoms. Such a limitation is necessary since otherwise, we could use a numerical feature that encodes the optimal value function $V^*(s)$, i.e., the perfect goal distance of all states $s$. With such a feature, the sketch rule $\{V^* > 0\} \mapsto \{V^*\downarrow\}$ makes all problems trivially solvable. Even with linear and quadratic features, we can capture complex state properties such as distances between objects.

## Proving Termination and Sketch Width

For each sketch introduced below we show that it uses goal-separating features, is terminating and has bounded and small sketch width. Showing that the features are goal separating is usually direct.

Proving **termination** is required to ensure that by iteratively moving from a state $s$ to a subgoal state $s' \in G_r(s)$ we cannot get trapped in a cycle. The conditions under which a sketch $R_\Phi$ is terminating are similar to those that ensure that a general policy $\pi_\Phi$ is terminating (Srivastava et al. 2011; Bonet and Geffner 2020b, 2021), and can be determined in polynomial time in the size of the sketch graph $G(\pi_\Phi)$ using the SIEVE procedure (Srivastava et al. 2011; Bonet and Geffner 2020b). Often, however, a simple syntactic procedure suffices that eliminates sketch rules, one after the other until none is left. This syntactic procedure is sound but not complete in general. In the following, we say that a rule $C \mapsto E$ *changes* a Boolean feature $b$ if $b \in C$ and $\neg b \in E$ or the other way around. The procedure iteratively applies one of the following cases until no rule is left (the sketch terminates) or until no further cases apply (there may be an infinite loop in the sketch): (a) eliminate a rule if it decreases a numerical feature $n$ ($n\downarrow$) that no other remaining rule can increase ($n\uparrow$ or $n?$); (b) eliminate a rule $r$ if it changes the value of a Boolean feature that no other remaining rule changes in the opposite direction; (c) mark all features that were used for eliminating a rule in (a) or (b) as these can only change finitely often; (d) remove rules $C \mapsto E$ that decrease a numerical feature $n$ or that change a Boolean feature $b$ to true (false) such that for all other remaining rules $C' \mapsto E'$ it holds that if $E'$ changes the feature in the opposite direction, i.e., $n\uparrow$, $n?$ or changes $b$ to false (true), there must be a condition on a variable in $C$ that is marked and is complementary to the one in $C'$, e.g., $n > 0 \in C$ and $n = 0 \in C'$ or $b \in C$ and $\neg b \in C'$.

Showing that a sketch for problem class $\mathcal{Q}$ has **sketch width** $k$ requires to prove that for all $R$-reachable states $s$ in all problem instances $P \in \mathcal{Q}$, the width of $P[s]$ is bounded by $k$. Remember that $P[s]$ is like $P$ but with initial state $s$, and goal states $G$ of $P$ combined with goal states $G_r(s)$ of all $r \in R$. The definition of $R$-reachability shows that we need a recursive proof strategy: informally, we show that (1) the feature conditions of a rule $r$ with nonempty subgoal $G_r(s)$ are true in all initial states $s$ of $\mathcal{Q}$, and (2) by following a rule, we land in a goal state or a state $s'$ where the feature conditions of another rule $r'$ with nonempty subgoal $G_{r'}(s')$ are true. To show that the sketch has width $k$, we prove that all subtasks $P[s]$ for traversed states $s$ have width $k$. We do this by providing an admissible chain $t_1, \ldots, t_m$ of size at most $k$ where all optimal plans for $t_m$ are also optimal plans for $P[s]$. We overapproximate the set of $R$-reachable states where necessary to make the proofs more compact. This implies that our results provide an upper bound on the actual sketch width but are sufficiently small. For space reasons, we give only one exemplary proof in the paper and provide the remaining proofs as well as a sketch for the Driverlog domain (Long and Fox 2003) in Drexler, Seipp, and Geffner (2021).

## Floortile

In the Floortile domain (Linares López, Celorrio, and Olaya 2015), a set of robots have to paint the tiles of a rectangular grid. There can be at most one robot $a$ on each tile $t$ at any time and the predicate $robot\text{-}at(a, t)$ is true iff $a$ is on tile $t$. If there is no robot on tile $t$ then $t$ is marked as clear, i.e., $clear(t)$ holds. Robots can move left, right, up or down, if the target tile is clear. Each robot $a$ is equipped with a brush that is configured to either paint in $black$ or $white$, e.g., $robot\text{-}has(a, black)$ is true iff the brush of robot $a$ is configured to paint in $black$. It is possible to change the color infinitely often. The goal is to paint a rectangular subset of the grid in chessboard style. If a tile $t$ has color $c$ then the predicate $painted(t, c)$ holds and additionally the tile is marked as not clear, i.e., $clear(t)$ does not hold. A robot $a$ can only paint tile $t$ if $a$ is on a tile $t'$ that is below or above $t$, i.e., $robot\text{-}at(a, t')$ holds, and $up(t', t)$ or $down(t', t)$ holds.

Consider the set of features $\Phi = \{g, v\}$ where $g$ counts the number of unpainted tiles that need to be painted and $v$ represents that the following condition is satisfied: for each tile $t_1$ that remains to be painted there exists a sequence of tiles $t_1, \ldots, t_n$ such that each $t_i$ with $i = 1, \ldots, n - 1$ remains to be painted, $t_n$ does not need to be painted, and for all pairs $t_{i-1}, t_i$ with $i = 2, \ldots, n$ holds that $t_i$ is above $t_{i-1}$, i.e., $up(t_{i-1}, t_i)$, or for all pairs $t_{i-1}, t_i$ with $i = 2, \ldots, n$ it holds that $t_i$ is below $t_{i-1}$, i.e., $down(t_{i-1}, t_i)$. Intuitively, $v$ is true iff a given state is solvable. The set of sketch rules $R_\Phi$ contains the single rule

$$r = \{v, g > 0\} \mapsto \{g\downarrow\}$$

which says that painting a tile such that the invariant $v$ remains satisfied is good.

**Theorem 2.** *The sketch for the Floortile domain is well-formed and has width* 2.

## TPP

In the Traveling Purchaser Problem (TPP) domain, there is a set of places that can either be markets or depots, a set of trucks, and a set of goods (Gerevini et al. 2009). The places are connected via a roads, allowing trucks to drive between them. If a truck $t$ is at place $p$, then atom $at(t, p)$ holds. Each market $p$ sells specific quantities of goods, e.g., atom $on\text{-}sale(g, p, 2)$ represents that market $p$ sells two quantities of good $g$. If there is a truck $t$ available at market $p$, it can buy a fraction of the available quantity of good $g$, making $g$ available to be loaded into $t$, while the quantity available at $p$ decreases accordingly, i.e., atom $on\text{-}sale(g, p, 1)$ and $ready\text{-}to\text{-}load(g, p, 1)$ hold afterwards. The trucks can unload the goods at any depot, effectively increasing the number of stored goods, e.g., atom $stored(g, 1)$ becomes false, and $stored(g, 2)$ becomes true, indicating that two quantities of good $g$ are stored. The goal is to store specific quantities of specific goods.

SIW fails in TPP because loading sufficiently many quantities of a single good can require buying and loading them from different markets. Making the goods available optimally requires taking the direct route to each market followed by buying the quantity of goods. Thus, the problem width is bounded by the number of quantities needed.

Consider the set of features $\Phi = \{u, w\}$ where $u$ is the number of goods not stored in any truck of which some quantity remains to be stored, and $w$ is the sum of quantities of goods that remain to be stored. The sketch rules in $R_\Phi$ are defined as:

$$r_1 = \{u > 0\} \mapsto \{u\downarrow\}$$
$$r_2 = \{w > 0\} \mapsto \{u?, w\downarrow\}$$

Rule $r_1$ says that loading any quantity of a good that remains to be stored is good. Rule $r_2$ says that storing any quantity of a good that remains to be stored is good.

**Theorem 3.** *The sketch for the TPP domain is well-formed and has width 1.*

*Proof.* The features are goal separating because $w = 0$ holds in state $s$ iff $s$ is a goal state. We show that the sketch $R_\Phi$ is terminating by iteratively eliminating rules: $r_2$ decreases the numerical feature $w$ which no other rule increments, so we eliminate $r_2$ and mark $w$. Now only $r_1$ remains and we can eliminate it since it decreases $u$, which is never incremented.

It remains to show that the sketch width is 1. Consider any TPP instance $P$. In the initial situation $s$, the feature conditions of at least one rule $r$ are true and the corresponding subgoal $G_r(s)$ is nonempty. Furthermore, whenever we use $r_1$ in some state $s$ to get to the next subgoal $G_{r_1}(s)$, we know that in this subgoal the feature conditions of $r_2$ must be true and its subgoal is nonempty, and it can be the case that the feature conditions of $r_1$ remain true and its subgoal is nonempty. At some point, the subgoal of $r_2$ is the overall goal of the problem. Next, regardless of which rule $r$ defines the closest subgoal $G_r(s)$ for an $R$-reachable state $s$, we show that $P[s]$ with subgoal $G_r(s)$ in $R$-reachable state $s$ has width 1.

We first consider rule $r_1$. Intuitively, we show that loading a good that is not yet loaded but of which some quantity remains to be stored in a depot has width at most 1. Consider states $S_1 \subseteq S$ where the feature conditions of $r_1$ are true, i.e., states where there is no truck that has a good $g$ loaded but of which some quantity remains to be stored in a depot. With $G_{r_1}(s)$ we denote the subgoal states of $r_1$ in $s \in S_1$, i.e., states where some quantity $q_l$ of $g$ is loaded into a truck $t$. The tuple $loaded(g, t, q_l)$ implies $G_{r_1}(s)$ in $s \in S_1$ in the admissible chain that consists of moving $t$ from its current place $p_1$ to the closest market $p_n$ that has $g$ available, ordered descendingly by their distance to $p_n$, buying $q_b$ quantities of $g$, loading $q_l$ quantities of $g$, i.e., $(at(t, p_1), \ldots, at(t, p_n), ready\text{-}to\text{-}load(g, p_n, q_b), loaded(g, t, q_l))$. Note that loading $q_l$ quantities can be achieved optimally by buying $q_b \geq q_l$ quantities.

Next, we consider rule $r_2$. Intuitively, we show that storing a good of which some quantity remains to be stored in a depot has width at most 1. Consider states $S_2 \subseteq S$ where the feature conditions of $r_2$ are true and some quantity of a good is loaded that remains to be stored, i.e., states where some quantity of a good $g$ remains to be stored in a depot, and some quantity $q_l$ of $g$ is loaded into a truck $t$ because it has width 1 (see above). With $G_{r_2}(s)$ we denote the subgoal states of $r_2$ in $s \in S_2$, i.e., states where the stored quantity $q_s$ of $g$ has increased from $q_s$ to $q_s'$ using a fraction of the loaded quantity $q_l' \leq q_l$. The tuple $stored(g, q_s')$ implies $G_{r_2}(s)$ in $s \in S_2$ in the admissible chain that consists of moving $t$ from its current place $p_1$ to the closest depot at place $p_n$, ordered descendingly by their distance to $p_n$, storing $q_l'$ quantities of $g$, i.e., $(at(t, p_1), \ldots, at(t, p_n), stored(g, q_s'))$.

We obtain sketch width 1 because all tuples in admissible chains have a size of at most 1. $\qquad\square$

## Barman

In the Barman domain (Linares López, Celorrio, and Olaya 2015), there is a set of shakers, a set of shots, and a set of dispensers where each dispenses a different ingredient. There are recipes of cocktails each consisting of two ingredients, e.g., the recipe for cocktail $c$ consists of ingredients $i_1, i_2$. The goal is to serve beverages, i.e., ingredients and/or cocktails. A beverage $b$ is served in shot $g$ if $g$ contains $b$. An ingredient $i$ can be filled into shot $g$ using one of the dispensers if $g$ is clean. Producing a cocktail $c$ with a shaker $t$ requires both ingredients $i_1, i_2$ of $c$ to be in $t$. In such a situation, shaking $t$ produces $c$. Pouring a cocktail from $t$ into shot $g$ requires $g$ to be clean. The barman robot has two hands which limits the number of shots and shakers it can hold at the same time. Therefore, the barman often has to put down an object before it can grasp a different object. For example, assume that the barman holds the shaker $t$ and some shot $g'$ and assume that ingredient $i$ must be filled into shot $g$. Then the barman has to put down either $t$ or $g'$ so that it can pick up $g$ with hand $h$. As in the Barman tasks from previous IPCs, we assume that there is only a single shaker and that it is initially empty.

Consider the set of features $\Phi = \{g, u, c_1, c_2\}$ where $g$ is the number of unserved beverages, $u$ is the number of

used shots, i.e., shots with a beverage different from the one mentioned in the goal, $c_1$ is true iff the first recipe ingredient of an unserved cocktail is in the shaker, and $c_2$ is true iff both recipe ingredients of an unserved cocktail are in the shaker. We define the following sketch rules for $R_\Phi$:

$$r_1 = \{\neg c_1\} \mapsto \{u?, c_1\},$$
$$r_2 = \{c_1, \neg c_2\} \mapsto \{u?, c_2\},$$
$$r_3 = \{u > 0\} \mapsto \{u{\downarrow}\},$$
$$r_4 = \{g > 0\} \mapsto \{g{\downarrow}, c_1?, c_2?\}.$$

Rule $r_1$ says that filling an ingredient into the shaker is good if this ingredient is mentioned in the first part of the recipe of an unserved cocktail. Rule $r_2$ says the same for the second ingredient, after the first ingredient has been added. Requiring the ingredients to be filled into the shaker in a fixed order ensures bounded width, even for arbitrary-sized recipes. Rule $r_3$ says that cleaning shots is good and rule $r_4$ says that serving an ingredient or cocktail is good.

**Theorem 4.** *The sketch for the Barman domain is well-formed and has width $2$.*

## Grid

In the Grid domain (McDermott 2000), a single robot operates in a grid-structured world. There are keys and locks distributed over the grid cells. The robot can move to a cell $c$ above, below, left or right of its current cell if $c$ does not contain a closed lock or another robot. The robot can drop, pick or exchange keys at its current cell and can only hold a single key $e$ at any time. Keys and locks have different shapes and the robot, holding a matching key, can open a lock when standing on a neighboring cell. The goal is to move keys to specific target locations that can be locked initially. Initially, it is possible to reach every lock for the unlock operation. SIW fails in this domain when goals need to be undone, i.e., a key has to be picked up from its target location to open a lock that is necessary for picking or moving a different key.

Consider the set of features $\Phi = \{l, k, o, t\}$ where $l$ is the number of locked grid cells, $k$ is the number of misplaced keys, $o$ is true iff the robot holds a key for which there is a closed lock, and $t$ is true iff the robot holds a key that must be placed at some target grid cell. We define the sketch rules for $R_\Phi$ as:

$$r_1 = \{l > 0\} \mapsto \{l{\downarrow}, k?, o?, t?\}$$
$$r_2 = \{l = 0, k > 0\} \mapsto \{k{\downarrow}, o?, t?\}$$
$$r_3 = \{l > 0, \neg o\} \mapsto \{o, t?\}$$
$$r_4 = \{l = 0, \neg t\} \mapsto \{o?, t\}$$

Rule $r_1$ says that unlocking grid cells is good. Rule $r_2$ says that placing a key at its target cell is good after opening all locks. Rule $r_3$ says that picking up a key that can be used to open a locked grid cell is good if there are locked grid cells. Rule $r_4$ says that picking up a misplaced key is good after opening all locks.

**Theorem 5.** *The sketch for the Grid domain is well-formed and has width $1$.*

## Childsnack

In the Childsnack domain (Vallati, Chrpa, and McCluskey 2018), there is a set of contents, a set of trays, a set of gluten-free breads, a set of regular breads that contain gluten, a set of gluten-allergic children, a set of children without gluten allergy, and a set of tables where the children sit. The goal is to serve the gluten-allergic children with sandwiches made of gluten-free bread and the non-allergic children with either type of sandwich.

The Childsnack domain has large bounded width because moving an empty tray is possible at any given time. The goal serialization fails because it gets trapped in deadends when serving non-allergic children with gluten-free sandwiches while leaving insufficiently many gluten-free sandwiches for the allergic children.

Consider the set of features $\Phi = \{c_g, c_r, s_g^k, s^k, s_g^t, s^t\}$ where $c_g$ is the number of unserved gluten-allergic children, $c_r$ is the number of unserved non-allergic children, $s_g^k$ holds iff there is a gluten-free sandwich in the kitchen, $s^k$ holds iff there is a any sandwich in the kitchen, $s_g^t$ holds iff there is a gluten-free sandwich on a tray, and $s^t$ holds iff there is any sandwich on a tray. We define the following sketch rules $R_\Phi$:

$$r_1 = \{c_g > 0, \neg s_g^k, \neg s_g^t\} \mapsto \{s_g^k, s^k\}$$
$$r_2 = \{c_g = 0, c_r > 0, \neg s^k, \neg s^t\} \mapsto \{s^k\}$$
$$r_3 = \{c_g > 0, s_g^k, \neg s_g^t\} \mapsto \{s_g^k?, s^k?, s_g^t, s^t\}$$
$$r_4 = \{c_g = 0, c_r > 0, s^k, \neg s^t\} \mapsto \{s_g^k?, s^k?, s_g^t?, s^t\}$$
$$r_5 = \{c_g > 0, s_g^t\} \mapsto \{c_g{\downarrow}, s_g^t?, s^t?\}$$
$$r_6 = \{c_g = 0, c_r > 0, s^t\} \mapsto \{c_r{\downarrow}, s_g^t?, s^t?\}$$

Rule $r_1$ says that making a gluten-free sandwich is good if there is an unserved gluten-allergic child and if there is no other gluten-free sandwich currently being served. Rule $r_2$ says the same thing for non-allergic children after all gluten-allergic children have been served and the sandwich to be made is not required to be gluten free. Rules $r_3$ and $r_4$ say that putting a gluten-free (resp. regular) sandwich from the kitchen onto a tray is good if there is none on a tray yet. Rule $r_5$ says that serving gluten-allergic children before non-allergic children is good if there is a gluten-free sandwich available on a tray. Rule $r_6$ says that serving non-allergic children afterwards is good.

**Theorem 6.** *The sketch for the Childsnack domain is well-formed and has width $1$.*

## Schedule

In the Schedule domain (Bacchus 2001), there is a set of objects that can have different values for the following attributes: shape, color, surface condition, and temperature. Also, there is a set of machines where each is capable of changing an attribute with the side effect that other attributes change as well. For example, rolling an object changes its shape to cylindrical and has the side effect that the color changes to uncolored, any surface condition is removed, and the object becomes hot. Often, there are multiple different

work steps for achieving a specific attribute of an object. For example, both rolling and lathing change an object's shape to cylindrical. But rolling makes the object hot, while lathing keeps its temperature cold. Some work steps are only possible if the object is cold. Multiple work steps can be scheduled to available machines, which sets the machine's status to occupied. All machines become available again after a single do-time-step action. The goal is to change the attributes of objects.

SIW fails in Schedule because it gets trapped into deadends when an object's temperature becomes hot, possibly blocking other required attribute changes. The following sketch uses this observation and defines an ordering over achieved attributes where first, the desired shapes are achieved, second, the desired surface conditions are achieved, and third, the desired colors are achieved.

Consider the set of features $\Phi = \{p_1, p_2, p_3, h, o\}$ where $p_1$ is the number of objects with wrong shape, $p_2$ is the number of objects with wrong surface condition, $p_3$ is the number of objects with wrong color, $h$ is the number of hot objects, and $o$ is true iff there is an object scheduled or a machine occupied. We define the following sketch rules $R_\Phi$:

$$r_1 = \{p_1 > 0\} \mapsto \{p_1\downarrow, p_2?, p_3?, o\}$$
$$r_2 = \{p_1 = 0, p_2 > 0\} \mapsto \{p_2\downarrow, p_3?, o\}$$
$$r_3 = \{p_1 = 0, p_2 = 0, p_3 > 0\} \mapsto \{p_3\downarrow, o\}$$
$$r_4 = \{o\} \mapsto \{\neg o\}$$

Rule $r_1$ says that achieving an object's goal shape is good. Rule $r_2$ says that achieving an object's goal surface condition is good after achieving all goal shapes. Rule $r_3$ says that achieving an object's goal color is good after achieving all goal shapes and surface conditions. Rule $r_4$ says that making objects and machines available is good. Note that $r_4$ does not decrease the sketch width but it decreases the search time by decreasing the search depth. Note also that $h$ never occurs in any rule because we want its value to remain constant.

**Theorem 7.** *The sketch for the Schedule domain is well-formed and has width 2.*

## Experiments

Even though the focus of our work is on proving polynomial runtime bounds for planning domains theoretically, we evaluate in this section how these runtime guarantees translate into practice. We implemented $SIW_R$ in the LAPKT planning system (Ramirez, Lipovetzky, and Muise 2015) and use the Lab toolkit (Seipp et al. 2017) for running experiments on Intel Xeon Gold 6130 CPU cores. For each planning domain, we use the tasks from previous IPCs. For each planner run, we limit time and memory by 30 minutes and 4 GiB. The benchmark set consists of a subset of tractable classical planning domains from the satisficing track of the 1998-2018 IPC where top goal serialization using SIW fails.

The main question we want to answer empirically is how much an SIW search benefits from using policy sketches. To this end, we compare $SIW(2)$ and $SIW_R(2)$ with the sketches for the planning domains introduced above. We use a width bound of $k=2$, since $SIW(k)$ and $SIW_R(k)$ are too slow to

compute in practice for larger values of $k$. We also include two state-of-the-art planners, LAMA (Richter and Westphal 2010) and Dual-BFWS (Lipovetzky and Geffner 2017a), to show that the planning tasks in our benchmark set are hard to solve with the strongest planners.

Table 1 shows results for the four planners. We see that the maximum effective width (MW) for $SIW_R(2)$ never exceeds the theoretical upper bounds we established in the previous section. For the domains with sketch width 2, the average effective width (AW) is always closer to 1 than to 2.

In the comparison we must keep in mind that $SIW_R$ is not a domain-independent planner as it uses a suitable sketch for each domain. $SIW(2)$ solves none of the instances in three domains (Barman, Childsnack, Floortile) because the problem width is too large. In the other four domains, it never solves more than half of the tasks. In contrast, $SIW_R(2)$ solves all tasks and is usually very fast. For example, $SIW(2)$ needs 74.7 seconds to solve the eleventh hardest TPP task, while $SIW_R(2)$ solves all 30 tasks in at most 0.4 seconds. This shows that our sketch rules capture useful information and that the sketch features are indeed cheap to compute.

Even with the caveats about planner comparisons in mind, the results from Table 1 show that policy sketches usually let $SIW_R$ solve the tasks from our benchmark set much faster than state-of-the-art domain-independent planners. The only exception is Schedule, where LAMA has a lower maximum runtime than $SIW_R$. The main bottleneck for $SIW_R$ in Schedule is generating successor states. Computing feature valuations on the other hand takes negligible time.

Overall, our results show that adding domain-specific knowledge in the form of sketches to a width-based planner allows it to solve whole problem domains very efficiently. This raises interesting research questions about whether we can learn sketches automatically to transform $SIW_R$ into a domain-independent planner that can reuse previously acquired information.

## Related Work

We showed that a bit of knowledge about the subgoal structure of a domain, expressed elegantly in the form of compact sketches, can go a long way in solving the instances of a domain efficiently, in provable polynomial time. There are other approaches for expressing domain control knowledge for planning in the literature, and we review them here.

The distinction between the actions that are "good" or "bad" in a fixed tractable domain can often be characterized explicitly. Indeed, the so-called **general policies**, unlike sketches, provide such a classification of all possible state transitions $(s, s')$ over the problems in $\mathcal{Q}$ (Francès, Bonet, and Geffner 2021), and ensure that the goals can always be reached by following any good transitions. Sketch rules have the same syntax as policy rules, but they do not constraint state transitions but define subgoals.

Logical approaches to domain control have been used to provide partial information about good and bad state transitions in terms of suitable formulas (Bacchus and Kabanza 2000; Kvarnström and Doherty 2000). For example, for the Schedule domain, one may have a formula in **linear temporal logic** (LTL) expressing that objects that need to be lathed

| Domain | SIW(2) | | | | SIW$_R$(2) | | | | LAMA | | Dual-BFWS | |
|---|---|---|---|---|---|---|---|---|---|---|---|---|
| | S | T | AW | MW | S | T | AW | MW | S | T | S | T |
| Barman (40) | 0 | – | – | – | 40 | 0.9 | 1.17 | 2 | 40 | 505.3 | 40 | 162.8 |
| Childsnack (20) | 0 | – | – | – | 20 | 10.8 | 1.00 | 1 | 6 | 2.6 | 8 | 216.9 |
| Driverlog (20) | 8 | 0.5 | 1.68 | 2 | 20 | 0.8 | 1.00 | 1 | 20 | 7.6 | 20 | 4.2 |
| Floortile (20) | 0 | – | – | – | 20 | 0.2 | 1.25 | 2 | 2 | 9.9 | 2 | 176.3 |
| Grid (5) | 1 | 0.1 | 2.00 | 2 | 5 | 0.1 | 1.00 | 1 | 5 | 3.6 | 5 | 3.7 |
| Schedule (150) | 62 | 1349.1 | 1.10 | 2 | 150 | 54.7 | 1.17 | 2 | 150 | 15.3 | 150 | 151.4 |
| TPP (30) | 11 | 74.7 | 2.00 | 2 | 30 | 0.4 | 1.00 | 1 | 30 | 16.5 | 29 | 99.6 |

Table 1: Comparison of SIW(2), SIW$_R$(2), the first iteration of LAMA, and Dual-BFWS. It shows the number of solved tasks (S), the maximum runtime in seconds for a successful run (T), the average effective width over all encountered subtasks (AW), and the maximum effective width over all encountered subtasks (MW).

and painted should not be painted in the next time step, since lathing removes the paint. This partial information about good and bad transitions can then be used by a forward-state search planner to heavily prune the state space. A key difference between these formulas and sketches is that sketch rules are not about state transitions but about subgoals, and hence they structure the search for plans in a different way, in certain cases ensuring a polynomial search.

Baier et al. (2008) combine control knowledge and preference formulas to improve search behavior and obtain plans of high quality, according to user preferences. The control knowledge is given in the Golog language and defines subgoals such that a planner has to fill in the missing parts. Since the control knowledge is compiled directly to PDDL, they are able to leverage off-the-shelve planners. The user preferences are encoded in an LTL-like language. Like our policy sketches, their approach can be applied to any domain. However, policy sketches aim at ensuring polynomial searches in tractable domains.

Hierarchical task networks or **HTNs** are used mainly for expressing general top-down strategies for solving classes of planning problems (Erol, Hendler, and Nau 1994; Nau et al. 2003; Georgievski and Aiello 2015). The domain knowledge is normally expressed in terms of a hierarchy of methods that have to be decomposed into primitive methods that cannot be decomposed any further. While the solution strategy expressed in HTNs does not have to be complete, it is often close to complete, as otherwise the search for decompositions easily becomes intractable. For this reason, crafting good and effective HTNs encodings is not easy. For example, the HTN formulation of the Barman domain in the 2020 Hierarchical Planning Competition (Höller et al. 2019) contains 10 high-level tasks (like *AchieveContainsShakerIngredient*), 11 primitive tasks (like *clean-shot*) and 22 methods (like *MakeAndPourCocktail*). In contrast, the PDDL version of Barman has only 12 action schemas, and the sketch above has 4 rules over 4 linear features. Note, however, that comparing different forms of control knowledge in terms of their compactness is not well-defined.

Finally, the need to represent the common subgoal structure of dynamic domains arises also in reinforcement learning (RL), where knowledge gained in the solution of some domain instances can be applied to speed up the learning of solutions to new instances of the same family of tasks (Finn, Abbeel, and Levine 2017). In recent work in deep RL (DRL) these representations, in the form of general **intrinsic reward functions** (Singh et al. 2010), are expected to be learned from suitable DRL architectures (Zheng et al. 2020). Sketches provide a convenient high-level alternative to describe common subgoal structures, but opposed to the related work in DRL, the policy sketches above are not learned but are written by hand. We leave the challenge of automatically learning sketches as future work and describe it briefly below.

## Conclusions and Future Work

We have shown that the language of policy sketches as introduced by Bonet and Geffner provides a simple, elegant, and powerful way for expressing the common subgoal structure of many planning domains. The SIW$_R$ algorithm can then solve these domains effectively, in provable polynomial time, where SIW fails either because the problems are not easily serializable in terms of the top goals or because some of the resulting subproblems have a high width. A big advantage of pure width-based algorithms like SIW and SIW$_R$ is that unlike other planning-based methods they can be used to plan with simulators in which the structure of states is available but the structure of actions is not.[2]

A logical next step in this line of work is to **learn sketches automatically**. In principle, methods similar to those used for learning general policies can be applied. These methods rely on using the state language (primitive PDDL predicates) for defining a large pool of Boolean and numerical features via a description logic grammar (Baader et al. 2003), from which the features $\Phi$ are selected and over which the general policies $\pi_\Phi$ are constructed (Francès, Bonet, and Geffner 2021). We have actually analyzed the features used in the sketches given above and have noticed that they can all be obtained from a feature pool generated in this way. A longer-term challenge is to learn the sketches automatically when using the same inputs as DRL algorithms, where there is no state representation language. Recent works that learn first-order symbolic languages from black box states or from states represented by images (Bonet and Geffner 2020a; Asai 2019) are important first steps in that direction.

---

[2] A minor difference then is that the novelty tests in IW($k$) are not exponential in $k - 1$ but in $k$.

## Acknowledgments

This work was partially supported by an ERC Advanced Grant (grant agreement no. 885107), by project TAILOR, funded by EU Horizon 2020 (grant agreement no. 952215), and by the Knut and Alice Wallenberg Foundation. Hector Geffner is a Wallenberg Guest Professor at Linköping University, Sweden. We used compute resources from the Swedish National Infrastructure for Computing (SNIC), partially funded by the Swedish Research Council through grant agreement no. 2018-05973.

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
