# OpenReview forum: "Expressing and Exploiting the Common Subgoal Structure of Classical Planning Domains Using Sketches"
_icaps-conference.org/ICAPS/2021/Workshop/HSDIP — HSDIP 2021_

### Official Review · AnonReviewer2 · 2021-05-27

**Confidence:** 4
**Overall Score:** Strong Accept

**Review:**

The paper follows up on previous work by Bonet and Geffner that introduced the notion of policy sketches. In this paper, it is shown that sketches can be manually designed for a number of domains: TPP, Barman, Floortile, Grid, Childsnack, Driverlog, and Schedule. The sketches are very simple, yet they provide the necessary domain-dependent information for a simple algorithm SIW to be able to solve all these domains in low polynomial time.

The paper is very convincing, showing that the policy sketches can be very useful even when compared with state of the art planners. Furthermore it is very well written, explaining the whole idea and the individual sketches in a detailed and comprehensive manner.



Some minor comments:

 - In TPP, there is an unstated assumption that the road network is strongly connected.

 - I did not follow this note: "Note that loading ql quantities can be achieved optimally by buying ql <= qb quantities.". What is this intending to say? That the sequence of actions just presented is not optimal? On a third read, I think this would be more clear if you just swap the last part, by buying qb >= ql quantities.

 - The description of the rules in Barman seems to be broken, missing r_w and with some sentences cut by half.

 - In Childsnack, shouldn't features s^k and s^t refer to all sandwiches, and not only regular sandwiches?  (the allergic children require gluten-free sandwiches but the rest can be served both kinds of sandwiches).


 - I am wondering about the order of the rules. I understand that this is just a set of rules and order does not matter for the algorithm. But it can still be relevant from the point of view of our understanding of what the rules are encoding. In this sense, I found very natural the order followed for Driverlog, placing first all rules that deal with p > 0, then with p=0, t > 0, and so on. This can almost be read as some decision tree, which checks the value of p, then of t, and so on. However the rules in Grid do not follow the same principle. To me, it'd be more natural to introduce r_3 before r_2, as r_2 will never be executed until l = 0.

  - In the comparison against HTN, I guess one needs to be careful when comparing the number of tasks/methods in HTN against the number of rules in the policy sketches. Policy sketches have moved a great deal of the complexity to the definition of the features itself, which could be any arbitrary procedure that runs in low polynomial time (but whose code could be quite complex compared to the definition of a method in HTN). This is not the case for most features used in this paper but, for example, the feature v in Floortile is quite complex to define. That said, I fully agree that the presented policy sketches are very compact and simple.


Minor:
  - page 3, then *the* value of f
  - page 4, sells two quantities (I'd use units instead)
  - page 6, possible *to* reach

---

> ### Author Response · Authors · 2021-05-31
> **Answers and comments**
>
> Dear AnonReviewer2,
>
> Thank you very much for your comments and questions.
>
> Regarding the assumption in TPP: without this assumption, we need to keep track of an invariant that checks that we do not reach an unsolvable state (comparable to what is done in the Floortile domain).
>
> Regarding buying qb and loading ql: indeed, using the expression qb >= ql is the correct way to write it. Thanks for providing us with the solution. Furthermore, we will use "units" instead of "quantities".
>
> Regarding Barman: thanks for spotting the broken paragraph. The fractional sentence should be commented out. Furthermore, the description of rule r2, which is currently missing, is: "Rule r2 says the same for the second ingredient after the first ingredient has been added."
>
> Regarding the order of rules: great suggestion! We will make use of this ordering strategy in future versions of this paper.
>
> Regarding the feature complexity: yes, that is correct. The features can hide some of the complexity of the sketches. We will elaborate on this in the comparison between policy sketches and HTN. The invariant in Floortile has the largest complexity among all features that we use. In many cases, the complexity is smaller than 10 and almost always smaller than 16, measured in the number of nodes in the abstract syntax tree over description logics constructors.

---

### Official Review · AnonReviewer1 · 2021-05-27
**Nice analysis of the complexity of solving several IPC domains with minimal knowledge provided in terms of policy sketches**

**Confidence:** 4
**Overall Score:** Accept

**Review:**

This paper introduces an in-depth analysis of the complexity of solving several standard IPC domains by using hand-crafted policy sketches.  Here, each sketch consists of a set of rules that take the form of a simple precondition and effect of boolean or numeric features. These rules separate the goals of a task such that the subgoals can be achieved sequentially by an SIW(k) search.

With this, the authors prove for a set of standard benchmark domains that very small sets of simple features are sufficient to solve the domain in polynomial time. The paper also shows an experimental evaluation comparing the new approach to standard SIW search, as well as to LAMA and DUAL-BFWS. The new variant compares very favourably, although the comparison is of course not really fair. Still, the authors show that very simply features, that can potentially be learned automatically, suffice to solve domains that are still challenging for state-of-the-art planners.

The authors also put their work in context, comparing to domain control knowledge provided, e.g., by LTL formulas or HTNs.

The paper is very well written and covers a topic interesting for HSDIP.

Very minor things:
- HTN could be spelled out once in the intro

---

### Decision · Program_Chairs · 2021-06-10

**Decision:**

Accept

**Comment:**

Both reviewers agreed that the paper should be accepted. We ask the authors to address the comments in the camera-ready version.